# Development of Novel Designs of Resistive Plate Chambers

**Burak Bilki** [1,2,3,*], **Yasar Onel** [2], **Jose Repond** [2], **Kutlu Kagan Sahbaz** [1,3,4], **Mehmet Tosun** [1,3,4] **and Lei Xia** [5] **on behalf of the CALICE Collaboration**

1 Department of Mathematics, Beykent University, Istanbul 34500, Turkey
2 Department of Physics and Astronomy, University of Iowa, Iowa City, IA 52242, USA
3 Turkish Accelerator and Radiation Laboratory, Ankara 06830, Turkey
4 AInstitute of Nuclear Sciences, Ankara University, Ankara 06100, Turkey
5 Argonne National Laboratory, Lemont, IL 60439, USA
* Correspondence: burak.bilki@cern.ch

**Abstract:** Resistive Plate Chambers (RPCs) are a key active media of the muon systems of current and future collider experiments as well as the CALICE (semi-)digital hadron calorimeter. The outstanding issues with RPCs can be listed as the loss of efficiency for the detection of particles when subjected to high particle fluxes and the limitations associated with the common RPC gases. We developed novel RPC designs with: low resistivity glass plates; a single resistive plate; and a single resistive plate and a special anode plate coated with high secondary electron emission yield material. The cosmic and beam tests confirmed the viability of these new approaches for calorimetric applications. The chambers also have improved single-particle response, such as a pad multiplicity close to unity. Here, we report on the construction of various different glass RPC designs and their performance measurements in laboratory tests and with particle beams. We also discuss future test plans, which include the long-term performance tests of the newly developed RPCs, investigation of minimal gas flow chambers, and feasibility study for the large-size chambers.

**Keywords:** resistive plate chambers; gaseous imaging and tracking detectors; hadron calorimetry

## 1. Introduction

Resistive Plate Chambers (RPCs) are particle detectors which were introduced in the 1980s [1] and have been widely used in High Energy Physics experiments since then. The experimental implementations are mostly on triggering and precision timing. They consist of two or more resistive plates of high resistance (glass or Bakelite) that are separated by thin gas gaps. The readout is provided either by strips or pads, which are placed on the outside of the chambers.

Under high particle fluxes, RPCs exhibit a significant loss of efficiency due to the high resistance of the resistive plates. Lower-resistivity glass samples are produced and purchased, and various size RPCs were constructed with them. As expected, lower-resistivity glasses offer higher rate capability. Although other types of low-resistivity planar materials are being probed as resistive plates, low-resistivity glass option remains viable due to large flexibility in tuning the glass composition and production parameters for optimal long-term performance.

In the context of studies of imaging calorimetry for a future lepton collider, as carried out by the CALICE Collaboration [2], we developed a novel design of RPC based on a single resistive plate. The RPCs were read out using the standard Digital Hadron Calorimeter (DHCAL) [3] electronic readout system featuring $1 \times 1$ cm$^2$ signal pads. Tests were performed with both cosmic rays and particle beams, and the results point towards an improvement in pad multiplicity and rate capability.

Recently, we developed 1-glass RPCs with anode planes coated with a thin layer of high secondary electron yield material. The purpose is to relax the requirements on the RPC

gases, both in terms of type and operational parameters, in order to mitigate the limitations of their use due to green house effects, which are present in many of them. A number of 1-glass RPC samples were produced and tested with both cosmic rays and particle beams resulting in promising first results.

Here, we report on the construction of various different glass RPC designs and their performance measurements in laboratory tests and with particle beams.

## 2. Development of Semi-Conductive Glass

One of the major areas that the RPCs require R&D for future implementations is the improvement of their rate capability. This has been a long-standing problem with limited solution thus far. The rate limitation is related to the usually high resistivity of the resistive plates used in their construction [4]. A simple approach to handle the limitation is to increase the electrical conductivity of the glass to allow the resistive plates to restabilize faster. Soda–lime silicate glasses in the current RPCs are well-known, inexpensive, and easily manufactured, on the other hand, they come in a bulk electrical conductivity at the order of $10^{-15}$ S/cm. For high-rate implementations, the target conductivity is two to five orders of magnitude higher. In addition to the conductivity requirements, the RPC glass must be homogenous, radiation-hard, and it must be easily manufactured and must not be ionically conductive to provide long-term stability.

In this context, we developed low resistivity vanadate-based glasses. The lead vanadates ended up with conductivities of the order of $10^{-10}$ S/cm, and the samples were not able to hold the high voltage with sparking across the plates. The target conductivity range was obtained with tellurium vanadates doped with zinc oxide (ZnO). Figure 1 (left) shows the electronic conductivity of the binary tellurium vanadate glass as a function of the vanadium oxide mole fraction. The conductivity of the binary system was between $10^{-6}$ and $10^{-10}$ S/cm and ZnO doping was an economical modifier to reduce the conductivity to within the target range. From 25% to 55% ZnO, conductivity ranged four orders of magnitude from around $10^{-11}$ to around $10^{-15}$ S/cm.

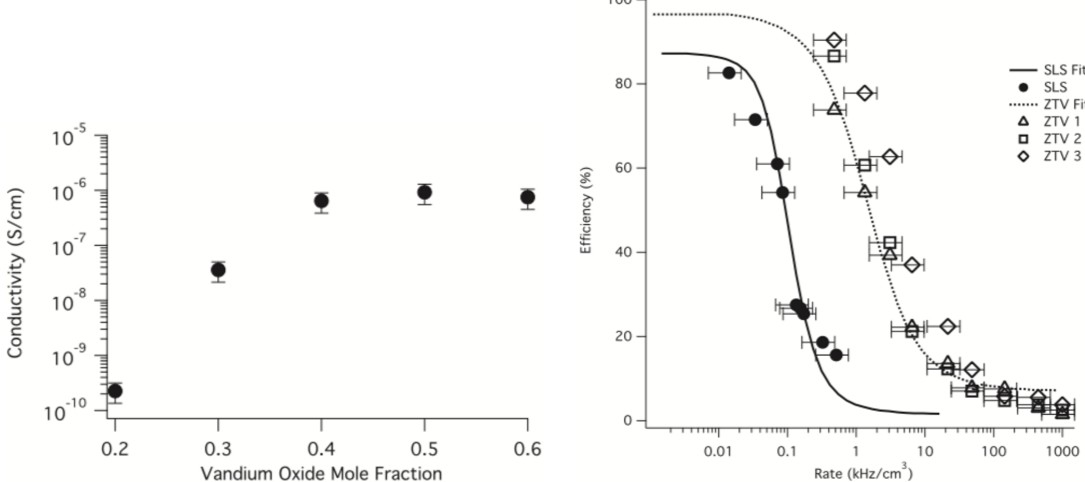

**Figure 1.** The electronic conductivity of the binary tellurium vanadate glass as a function of the vanadium oxide mole fraction (**left**) and the efficiency vs. particle rate for the RPCs made with zinc-tellurium vanadate glasses (**right**).

The glass composition of $0.40ZnO–0.40TeO_2–0.20V_2O_5$ was used to make three 5 cm $\times$ 5 cm two-glass RPCs. The RPCs were tested at the Fermilab Test Beam Facility (FTBF) [5] with 120 GeV proton beam. Figure 1 (right) shows the efficiency vs. particle rate for the RPC made with soda–lime silicate glasses (denoted as SLS) and the three RPCs made with the zinc tellurium vanadate glasses (denoted ZTV1, ZTV2 and ZTV3). The improvement in the rate capability of the RPCs at a given efficiency is more than an order

of magnitude [6]. Since the production is made in house, the conductivity can be tuned to very fine levels. The performance of the RPCs is consistent across different glass production campaigns and is also consistent with the theoretical calculations [4] denoted with the solid curves. The final composition can be easily transferred to larger production facilities. On the other hand, the mechanical properties of several square-meter-size glasses with this composition should be studied in detail once available.

### 3. Development of 1-Glass RPC

The novel 1-glass RPC design offers a number of distinct advantages including an average pad multiplicity close to unity, indicating better position resolution and easier calibration; less strict parameters on the resistive layer of the cathode plate, as the performance of the two- (or multi-) glass chamber mostly depends on the quality of the coating on the anode plate; lower thickness, which is highly desirable for calorimetry; and an improved rate capability [7].

Chambers with a lateral size of $32 \times 48$ cm$^2$ were built with a single soda–lime float glass and were read out with standard DHCAL electronics. Figure 2 shows pictures of the readout (left) and front (right) side of the 1-glass RPC. The average efficiency of the chambers was 95%, and the pad multiplicity was close to unity as measured in the cosmic ray test stand [7].

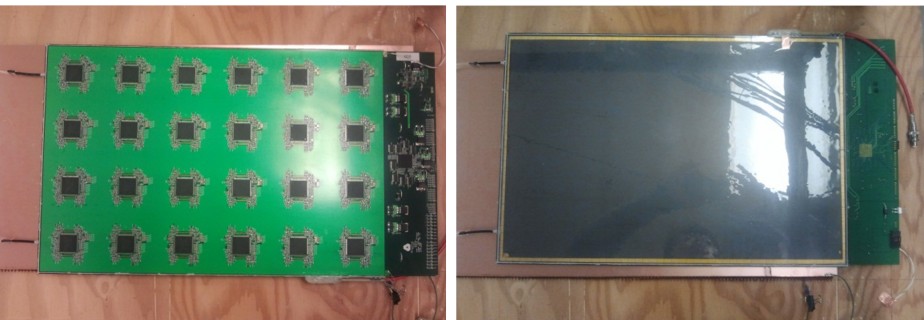

**Figure 2.** Pictures of the readout (**left**) and front (**right**) side of the 1-glass RPC. The readout side picture depicts the array of the DCAL chips and the data concentrator circuitry of the digital readout. The front side picture depicts the cathode glass covering the entire area of the 1 cm × 1 cm pads.

### 4. Measurement of Rate Capability

Three different RPC designs were tested for rate capability at FTBF:

- 2-glass RPCs with standard glass: The chambers were built with two standard soda–lime float glass plates with a thickness of 1.1 mm each. The gas gap was 1.2 mm. The chambers were $20 \times 20$ cm$^2$ in size.

- 1-glass RPCs with standard glass: The chambers were built with one standard soda–lime float glass plate with a thickness of 1.15 mm. The gas gap was also 1.15 mm. The lateral size of the chamber was dictated by the size of the readout board, i.e., $32 \times 48$ cm$^2$, as described in the previous section.

- 2-glass RPCs with semi-conductive glass: These chambers utilize 1.4 mm semi-conductive glass with a bulk resistivity several orders of magnitude smaller than standard soda–lime float glass, obtained from Schott Glass Technologies Inc. [8]. The gas gap of these chambers was also 1.15 mm and the area of the chambers measured $20 \times 20$ cm$^2$.

The usually pencil-like 120 GeV primary proton beam of FTBF was defocused upstream of the experimental hall, and the Gaussian beam profile was measured with the wire chambers. The widths in the $x$ and $y$ directions were measured to be $\sigma_x = 1.0$ cm and $\sigma_y = 0.8$ cm. In the calculation of the beam intensity, in units of Hz/cm$^2$, the size of the beam spot was taken to be $2\sigma_x \times 2\sigma_y$, with an error derived from the measurement error of the widths of the Gaussians.

Figure 3 shows the efficiency (left) and average pad multiplicity (center) as a function of beam rate for six different RPCs [9]. The performance across different construction campaigns is consistent, and the rate at which the chambers retain high efficiency is observed to increase with decreasing overall resistance of the chambers. The 1-glass RPC is approximately a factor of two better than the two-glass RPC in terms of rate capability. The average pad multiplicity is close to unity for the entire range of particle rates for the 1-glass chambers. For the 2-glass chambers, the average pad multiplicity remains below two. Figure 3 (right) shows the rate at 50% efficiency as a function of conductance per area of the glass plates. The points are fit empirically to the following function which was drawn as a red curve: $I_{50\%} = 1.7 \times 10^5 + 3.2 \times 10^6 H - 1.7 \times 10^8 H^3$ where $H = 1/\log_{10} G$, $G$ being the conductance per unit area of the glass plates. There is indication that higher rate capability can be achieved with further R&D.

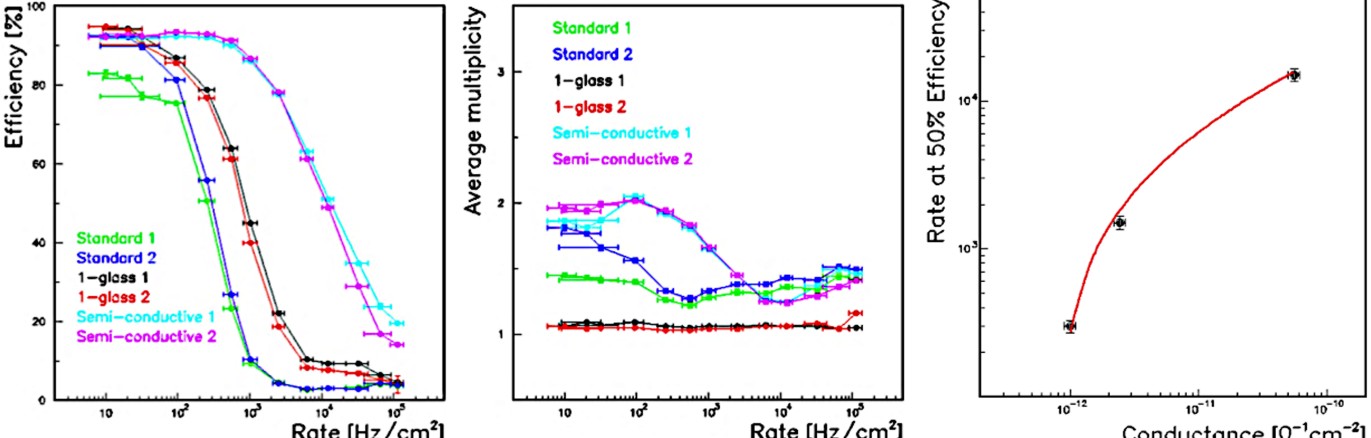

**Figure 3.** The efficiency (**left**) and the average pad multiplicity (**center**) as a function of beam rate for six different RPCs, and the rate at 50% efficiency as a function of conductance per area of the glass plates (**right**).

## 5. Development of Hybrid RPC

In order to mitigate the performance limitations associated with the alternative RPC gas mixtures, reduce the overall fresh gas needs also investigating the possibility of sealed chambers, and study the alternative multiplication mechanisms, we probed a new technique based on further electron multiplication in a thin layer of high secondary electron yield material. The first set of materials probed were $Al_2O_3$ and $TiO_2$. The material is applied as a coating on the anode plane, which is inside the chamber for the 1-glass RPC designs. The coating of $Al_2O_3$ was carried out with magnetron sputtering, and the coating of $TiO_2$ was applied with an airbrush. We constructed several 10 cm × 10 cm chambers, also standard 1- and 2-glass RPCs. Following the laboratory tests, the RPCs were tested with FTBF muons. The list of the RPCs tested is as follows: one standard 1-glass RPC; one standard 2-glass RPC; two 1-glass RPCs with anodes coated with 500 nm and 350 nm $Al_2O_3$ ($Al_2O_3$_v1 and $Al_2O_3$_v2); and three 1-glass RPCs with anodes coated with 1 mg/cm$^2$, 0.5 mg/cm$^2$, and 0.15 mg/cm$^2$ $TiO_2$ ($TiO_2$_v1, $TiO_2$_v2, and $TiO_2$_v3). The gas mixture was the standard DHCAL mixture: R134a (94.5%), Isobutane (5.0%), and $SF_6$ (0.5%); the gas flow rate was 2–3 cc/min, roughly half of the rate for the DHCAL.

Figure 4 shows the efficiency as a function of the applied high voltage. The standard 2-glass RPC becomes 90% efficient around 8.5 kV. Similarly, if one considers the high voltage value at the 90% efficiency crossing as a measure, for the standard 1-glass RPC and $TiO_2$_v3, it is around 7.5 kV. The advantage of the 1-glass RPC over the 2-glass RPC is manifest, and there is no measurable effect on the performance with 0.15 mg/cm$^2$ $TiO_2$ coating applied on the anode plate.

The major improvement on the performance is obtained with the RPCs which have anode plates coated with $Al_2O_3$ and thicker $TiO_2$. The 90% efficiency crossing voltage

setting is around 6.5 kV, clearly indicating the contribution of the electron multiplication in the coating since the standard 1-glass RPC efficiency is negligible at these voltages. Following the promising results with these first-generation hybrid RPCs, data analysis and simulation studies to quantify the effect of the coating as a function of the material and its thickness are underway, and further tests with segmented anode and different gases are planned.

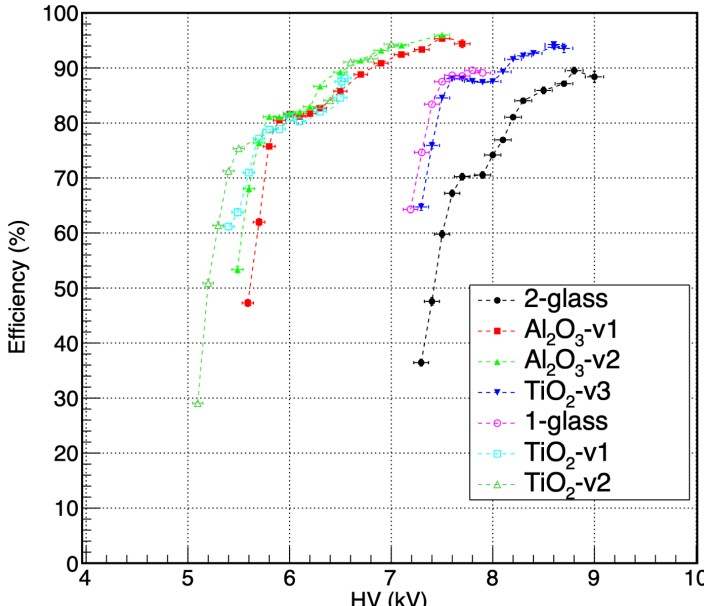

**Figure 4.** The efficiency as a function of applied high voltage for seven RPCs, including the five hybrid RPCs.

## 6. Conclusions

Novel designs of Resistive Plate Chambers were developed in order to respond to several outstanding issues.

A dedicated R&D was performed to develop semi-conductive glasses with flexible design parameters. It was found that the $0.40ZnO–0.40TeO_2–0.20V_2O_5$ glass provides a feasible starting point with a large phase space of options of constituents and processes to pursue further R&D.

The 1-glass RPC design offers several advantages over 2-glass RPCs, such as an average pad multiplicity around one and an increased rate capability. It also starts a new chapter where the in-chamber anode plate can be made more functional. By coating the anode plate with high secondary electron yield materials, electron multiplication in the chamber can be enhanced considerably. R&D is underway to fully characterize the newly developed, so-called hybrid RPCs. The hybrid RPCs have the potential to mitigate the limitations associated with RPC gases and to relax the overall operating conditions.

Several RPCs, including 1-glass and 2-glass RPCs made with the standard soda–lime float glass, and 2-glass RPCs made with semiconductive glasses, were tested for their rate capabilities in a 120 GeV proton beam at FTBF. The results indicate that with increasing conductance per area of the glass plates, the rate capability of RPCs increases. In addition, the range of particle rates for which the chambers retain their full particle detection efficiency also increases. An empirical relation for the dependence of the rate at 50% efficiency on the conductance per unit area of the glass plates was obtained.

**Author Contributions:** Investigation, K.K.S. and M.T.; Project administration, B.B. and J.R. and L.X.; Writing–review & editing, Y.O. All authors have read and agreed to the published version of the manuscript.

**Funding:** This research received no external funding.

**Data Availability Statement:** Not applicable.

**Acknowledgments:** B. Bilki, K. K. Sahbaz and M. Tosun acknowledge support under Tübitak grant no 118C224.

**Conflicts of Interest:** The authors declare no conflict of interest.

## References and Note

1.  Santonico, R.; Cardarelli, R. Development of resistive plate counters. *Nucl. Instr. Meth.* **1981**, *187*, 377. [CrossRef]
2.  Available online: https://twiki.cern.ch/twiki/bin/view/CALICE/WebHome (accessed on 31 August 2022).
3.  Adams, C.; Bambaugh, A.; Bilki, B.; Butler, J.; Corriveau, F.; Cundiff, T.; Drake, G.; Francis, K.; Furst, B.; Guarino, V.; et al. Design, construction and commissioning of the Digital Hadron Calorimeter—DHCAL. *J. Instrum.* **2016**, *11*, P07007. [CrossRef]
4.  Bilki, B.; Butler, J.; May, E.; Mavromanolakis, G.; Norbeck, E.; Repond, J.; Underwood, D.; Xia, L.; Zhang, Q. Measurement of the rate capability of Resistive Plate Chambers. *J. Instrum.* **2009**, *4*, P06003. [CrossRef]
5.  Available online: https://ftbf.fnal.gov/ (accessed on 31 August 2022).
6.  Johnson, N.; Wehr, G.; Hoar, E.; Xian, S.; Akgun, U.; Feller, S.; Affatigato, M.; Repond, J.; Xia, L.; Bilki, B. Electronically Conductive Vanadate Glasses for Resistive Plate Chamber Particle Detectors. *Int. J. Appl. Glass Sci.* **2015**, *6*, 26. [CrossRef]
7.  Bilki, B.; Corriveau, F.; Freund, B.; Neubüser, C.; Onel, Y.; Repond, J.; Schlereth, J.; Xia, L. Tests of a novel design of Resistive Plate Chambers. *J. Instrum.* **2015**, *10*, P05003. [CrossRef]
8.  Schott GlassTechnologies Inc. 400 York Ave, Duryea, PA 18642, USA.
9.  Affatigato, M.; Akgun, U.; Bilki, B.; Corriveau, F.; Freund, B.; Johnson, N.; Neubüser, C.; Onel, Y.; Repond, J.; Xia, L. Measurements of the rate capability of various Resistive Plate Chambers. *J. Instrum.* **2015**, *10*, P10037. [CrossRef]