# Peer review of "Development of Novel Designs of Resistive Plate Chambers"

_instruments, doi:10.3390/instruments6030035_

Round 1
Reviewer 1 Report
Dear Authors,
the paper is overall in good shape, I have only a few comments:
Figure 1 a/ lines 56 following: The conductivity ranges mentioned in the text and shown in Figure 1 left do not match up. You are either showing the wrong figure here or you need to correct the text.
line 66: In house production: You could maybe comment on how easy it would be to transfer the production to a commercial supplier for larger scale productions.
line 68/ Figure 1 b: the text says "theoretical calculations", the plot (Figure 1 right) says fit - which one is it ? Please clarify in the paper. If it would be a calculation without direct input of the data points, it would be quite impressive.
Section 3 / e.g. lines 79: The average efficiency and pad multiplicity of the chambers quoted here - should we refer to Figure 3 for the actual measurement or do you have maybe some previous work to cite here or a dedicated measurement plot to look at?
line 110: Another fit function - did you chose this functional form because it fits the data well or what is its motivation ? Please comment in the paper. I assume the three numerical parameters are all free fit parameters ? Given that you fit a three parameter function through three points I doubt you can learn much from it. Although I agree with the general trend of the data and the conclusion that even higher rate capability could be achieved with more R&D
As with MDPI's policies a paper can be accepted even if reviewer(s) ask for minor changes, I would like to suggest that you apply the above comments in that case at the proof stage.
All the best
Author Response
On behalf of the authors, I would like to thank the reviewer for the careful reading and constructive comments and suggestions. Please find my replies below:
==========
Dear Authors,
the paper is overall in good shape, I have only a few comments:
Figure 1 a/ lines 56 following: The conductivity ranges mentioned in the text and shown in Figure 1 left do not match up. You are either showing the wrong figure here or you need to correct the text.
========== A bit more detail was needed in the text. Added.
line 66: In house production: You could maybe comment on how easy it would be to transfer the production to a commercial supplier for larger scale productions.
========== Done.
line 68/ Figure 1 b: the text says "theoretical calculations", the plot (Figure 1 right) says fit - which one is it ? Please clarify in the paper. If it would be a calculation without direct input of the data points, it would be quite impressive.
========== These curves are actually the results of theoretical calculations presented in B. Bilki et al., "Measurement of the rate capability of Resistive Plate Chambers", JINST 4, P06003, 2009. The only free parameter is the turn-on voltage which cannot be measured for the case of digital readout. This voltage value is predicted based on the data points.
Section 3 / e.g. lines 79: The average efficiency and pad multiplicity of the chambers quoted here - should we refer to Figure 3 for the actual measurement or do you have maybe some previous work to cite here or a dedicated measurement plot to look at?
========== Added a reference to B. Bilki et al., "Tests of a novel design of Resistive Plate Chambers", JINST 10, P05003, 2015.
line 110: Another fit function - did you chose this functional form because it fits the data well or what is its motivation ? Please comment in the paper. I assume the three numerical parameters are all free fit parameters ? Given that you fit a three parameter function through three points I doubt you can learn much from it. Although I agree with the general trend of the data and the conclusion that even higher rate capability could be achieved with more R&D
========== This was an empirical fit which was also mentioned in M. Affatigato et al., "Measurements of the rate capability of various Resistive Plate Chambers", JINST 10, P10037, 2015. This information was added.
As with MDPI's policies a paper can be accepted even if reviewer(s) ask for minor changes, I would like to suggest that you apply the above comments in that case at the proof stage.
All the best
========== All comments were highly valuable and constructive, so all of them were implemented immediately.
Reviewer 2 Report
This proceeding describes the development and testing of several configurations of Resistive Plate Chambers (RPCs) with a particular focus on low resistivity RPCs as especially needed for RPC-based hadronic calorimeters in addition to the more traditional use in muon detectors. The development of these detector technologies is important for the future of particle detectors and this work is well motivated. I have just a handful of comments for the authors on ways to improve the manuscript prior to publication.
* I find this manuscript to be fairly well written, although there are a number of places where the English could be improved. I do think the manuscript quality could be significantly improved with more direct, concise language.
* L1: "RPCs are the key active media..." -> "RPCs are a key active media..."? "The" seems like a much too strong statement generally.
* Figure 2. The caption, labels, and/or the text could help improve this figure. It's not clear to me what the reader should learn from this series of grey squares. More detailed descriptions would be helpful.
* L110: I'm struggling to understand the motivation for the fit function here. It's a third order polynomial in 1/log(x) with the second order term omitted, and there's no discussion of this strange form. What's even stranger is that this is a three parameter fit that is attempting to fit three points, and yet the fit function doesn't seem to go through all three points. Is there not a simpler function that would actually describe the data well?
* Figure 3: At least on my printed copy, the quality of these figures is much lower than the others in the manuscript.
* There is a space before every percent sign that shouldn't be there.
* L148: "...provides an outperforming starting point with unbound options..." is very clumsy.
Author Response
On behalf of the authors, I would like to thank the reviewer for the careful reading and constructive comments and suggestions. Please find my replies below:
==========
This proceeding describes the development and testing of several configurations of Resistive Plate Chambers (RPCs) with a particular focus on low resistivity RPCs as especially needed for RPC-based hadronic calorimeters in addition to the more traditional use in muon detectors. The development of these detector technologies is important for the future of particle detectors and this work is well motivated. I have just a handful of comments for the authors on ways to improve the manuscript prior to publication.
* I find this manuscript to be fairly well written, although there are a number of places where the English could be improved. I do think the manuscript quality could be significantly improved with more direct, concise language.
========== I did another proof-reading and made improvements at a few occasions.
* L1: "RPCs are the key active media..." -> "RPCs are a key active media..."? "The" seems like a much too strong statement generally.
========== Done.
* Figure 2. The caption, labels, and/or the text could help improve this figure. It's not clear to me what the reader should learn from this series of grey squares. More detailed descriptions would be helpful.
========== Improved the caption.
* L110: I'm struggling to understand the motivation for the fit function here. It's a third order polynomial in 1/log(x) with the second order term omitted, and there's no discussion of this strange form. What's even stranger is that this is a three parameter fit that is attempting to fit three points, and yet the fit function doesn't seem to go through all three points. Is there not a simpler function that would actually describe the data well?
========== This was an empirical fit and this information is added.
* Figure 3: At least on my printed copy, the quality of these figures is much lower than the others in the manuscript.
========== Tried to improve a bit.
* There is a space before every percent sign that shouldn't be there.
========== Done.
* L148: "...provides an outperforming starting point with unbound options..." is very clumsy.
========== Fixed.